# Understanding Contextual Factors Effects and Their Implications for Italian Physiotherapists: Findings from a National Cross-Sectional Study

**DOI:** 10.3390/healthcare9060689

**Published:** 2021-06-07

**Authors:** Mattia Bisconti, Davide Venturin, Alessandra Bianco, Valentina Capurso, Giuseppe Giovannico

**Affiliations:** 1Department of Medicine and Health Science “Vincenzo Tiberio”, University of Molise, c/o Cardarelli Hospital, C/da Tappino, 86100 Campobasso, Italy; mattia.bisconti@unimol.it (M.B.); ft.alessandrabianco@gmail.com (A.B.); giuseppegiovannico73@gmail.com (G.G.); 2Physiotherapy and Manual Therapy, Physiotherapy Department UPMC Italy Salvator Mundi International Hospital, Viale delle Mura Gianicolensi 67, 00152 Rome, Italy; 3Kinè Physiotherapic Center, 31020 San Vendemiano, Italy; 4Reha Medica, Centro Medico di Riabilitazione, 70014 Conversano, Italy; 5Poliambulatorio San Luca, 70023 Gioia del Colle, Italy; valentinacapurso.12@gmail.com

**Keywords:** contextual factors, context, pain management, Italian physiotherapists, placebo effects, nocebo effects

## Abstract

An online cross-sectional survey was conducted using Google Docs software. The aim was to understand the management of contextual factors and to identify which are most relevant and which clinicians underestimate. A total of 1250 physiotherapists were chosen from the database of the Manual Therapists group mailing list (GTM-IFOMPT MO) from July to August 2020. A total of 699 responses were received that were considered valid (56%). Participants (40.83%) identified contextual factors (CFs) as “any element, even involuntary, with which the patient interacts during treatment”. Physiotherapists individually chose the representation of CF with the “therapeutic relationship” (82.9%), followed by “therapeutic setting” (75.8%). This choice differed between participants belonging to different age groups. Participants favor communication strategies (76.93%). More than half (57.88%) pay attention to patient involvement during the course of care; and in response to the patients’ doubts about the use of treatments with limited scientific efficacy, they suggest different medical treatments. The patient’s previous clinical experience is not considered significant and does not influence the choice of treatment. Subsequently, however, the participants reported that they stimulate the patients’ positive expectations of the success of the clinical outcome (45.27%). Knowledge of contextual factors in physiotherapy appears limited and very heterogeneous. Future research could increase the focus on professional development.

## 1. Introduction

Many reviews [1,2,3,4] and questionnaires [5,6,7,8] have highlighted the manner in which the therapeutic outcome is strongly influenced by the entire atmosphere surrounding the meeting between patient and healthcare professional [2,4,9]. The clinical condition of patients may change along with the treatment, worsening or improving without a particular expected evident reason [9]. This clinical phenomenon, in the current state of research, seems to be determined not only by the adequacy of the therapy adopted, but also by the way it is delivered [9]. The therapeutic arena, so called by Rossettini G., includes several elements such as therapeutic signs and rituals, the encounter between patient and physiotherapist, verbal and non-verbal interaction, the healthcare professional’s appearance and behaviors, and patient expectations [9,10]. These features play a significant role during each clinical stage and are called contextual factors (CFs) [4,9]. By definition, CFs are all those physical, psychological, and social elements that distinguish any health encounter [1]. In physiotherapy, there are five CF categories: *physiotherapist characteristics* (i.e., professional reputation, appearance, beliefs, behaviors); *patient’s characteristics* (i.e., expectations, preferences, previous experiences, musculoskeletal condition, gender, age); *patient–physiotherapist relationship* (i.e., verbal and non-verbal communication, therapeutic alliance); *treatment* (i.e., clear diagnosis, overt therapy, patient-centered approach, therapeutic touch), *therapeutic environment* (i.e., architecture, interior design, equipment) [9,11].

Patients’ exposure to a positive environment with positive CFs causes a placebo effect, resulting in improvement of symptoms (i.e., analgesia) [9]. A negative context, by contrast, could engender a nocebo effect, causing a worsening of the condition itself (i.e., hyperalgesia) [9]. The CFs affect not just pain, they also have a crucial role in motor performance and on overall patient satisfaction [5,11]. At the neurobiological level, CFs stimulate specific neurotransmitters (i.e., cholecystokinin, opioids, endocannabinoids, vasopressin and dopamine) that are responsible for significant clinical effects [5]. Furthermore, CFs can influence patients in different ways: the context responsiveness seems not to be continuous, but depends on the specific characteristics of the situation [6]. The patient is not always responsive to certain context elements: the same person, in similar contexts but different moments of their life, could respond in different ways [9].

In this scenario, we hypothesize that the distinct clinical success between two different healthcare professionals may depend on their ability to integrate contextual factors into clinical practice, while also both working according to scientific evidence and applying clinical guidelines. The available literature has reported the use of placebo effects in specific groups of healthcare workers in Europe, America and the Middle East. The literature documents their application by 17% and 80% of doctors, by 51% and 100% of nurses and by 52% of Italian OMPTs [5]. In contrast, the CFs knowledge and application by Italian physiotherapists are not yet available. The questionnaire survey aims, therefore, to understand how well the participants know the CFs, how often they use them and which CFs they prefer to apply. At this moment in history, such information should perforce belong to the clinicians’ background to increase the success of their interventions.

This cross-sectional study starts from a literary review whose reference databases for the search strategy were PubMed, PEDro and Google scholar. A manual search was also performed, and only studies available in full-text in English were included. The strategy used for the literature research involved the combination of the following keywords and MESH terms: ‹*Social Norms*›, ‹*Cultural Competency*›, ‹*Primary Health Care*›, ‹*Placebo*›, ‹*Nocebo*›, ‹*Environment*›, ‹*Musculoskeletal Pain*›, ‹*Pain Management*›, ‹*Chronic Pain*›, ‹*Acute Pain*›, ‹*Pain Perception*›, ‹*Physiotherapy*›, ‹*Clinical practice*›, ‹*Contextual factors*›. Research has shown that in the last two decades, these issues have undergone a strong re-evaluation; however, many studies have focused on general medical practice, while few had a specific physiotherapy focus.

## 2. Materials and Methods

### 2.1. Design and Ethical Issues

Our research team issued a nationwide cross-cutting web-based survey following the Checklist for Reporting Results of Internet E-surveys (CHERRIES) [12] and Strengthening the Reporting of Observational Studies in Epidemiology (STROBE) [13]. It is a result of a project aimed at investigating the knowledge and governance of contextual factors (CFs) in clinical practice by Italian physiotherapists.

### 2.2. Participants and Setting

All respondents were Italian physiotherapists. The participation was voluntary, without compensation. A valid e-mail address and a Bachelor in physiotherapy were mandatory to complete the survey; it was not necessary to have any specialization. Respondents were required to be working, as a private physiotherapist or in a public healthcare service, in Italy when we made survey disclosure.

### 2.3. Instrument Questionnaire Development

To improve the validity, formulation and clarity of the contents, two physiotherapists modified and implemented the first list of elaborated questions. They are lecturers at the University of Molise, and they are experts in musculoskeletal disorder rehabilitation and OMPT. During the preparation of the questionnaire, we pointed out the considered variables in order to clearly explain the suggested topics, the reason for the survey and the goals we aimed to achieve. Subsequently, we showed the paper to a Master of Science in Statistical Sciences, who further verified the reliability of the psychometric requirements in order to protect the accuracy of the obtained data before releasing the questionnaire. Compared to previous questionnaires with a similar purpose, in our survey, for the first time, each context factor (currently acknowledged) is individually studied [5,6,7,8]. The aim is to understand how to manage and identify the most relevant and underestimated CFs. Additionally, we focused on the region of Molise and regional Italian universities in order to determine any connection between CFs and education. 

Once the preliminary version was ready, ten people received the questionnaire. They belonged to the determined target group, but they were not part of the project (seven highly experienced physiotherapists in clinical practice, two clinical physiotherapists and experts in survey design, and one statistician). These experts worked independently, identifying confusing or unclear questions and testing the length of the questionnaire [14]. Corrections were progressively included, taking into account their feedback.

The final version consisted of 25 critically evaluated questions for *face and content validity* [15]. Before they answered the questions, the system notified respondents that responses were anonymous, in accordance with the Legislative Decree 196/2003. The entry data were anonymous. We did not collect sensitive personal data. The attendees could agree to the use of their answers by the researchers only (by pressing the “next” button).

The questionnaire’s preliminary page focuses on the presentation and methodological premise. Here we introduce the subject matter, name of the authors, estimated compiling time (about 7 min) and purpose of the study.

The first section (section A) of the questionnaire gathers the sample’s socio-demographic information. It consists of eight questions: five multiple-choice questions and three open questions. Section B explores the respondents’ first approach to the topic. The additional sections (n = 7) are composed of multiple-choice questions. They have a logical sequence, and an initial title that indicates the context factor characteristic being studied in that section. The last one seeks to understand the respondents’ opinions about delving into the subject matter within the context of the university training path. The response could be dichotomous (yes/ no). Google Forms (https://www.google.it/intl/it/forms/about/) (accessed on 30 June 2020) was the chosen platform for data collection and online management. The complete original questionnaire is attached.

### 2.4. Data Collection Instrument and Process

Our team released the questionnaire on 1 July 2020, and it was available until August 24, 2020. The participants received a Google Forms invitation link (see Appendix A). The team shared the link with 1250 physiotherapists who were Group of Manual Therapists mailing list subscribers (GTM). An introductory message to the link described the growing interest in the topic.

### 2.5. Data Analysis

Google Modules automatically imported the respondents’ answers into an Excel document immediately after completion of the survey. The system deleted incomplete answers and duplicates. R codes detected obvious weak responses [16]. Google Forms saved data in the format .xlsx.

For questions with one answer, our team applied descriptive statistics: mathematical average and standard deviation (SD). The software statistically calculated continuous variables and confidence intervals (CI) 95%. The system assigned frequencies and percentages to dichotomous, nominal and ordinal variables.

For questions with multiple-choice answers, the software calculated the absolute and relative frequencies for each answer combination. The Cramer V index [17] examined the association between individual characteristics (section A) and single-choice responses (sections B, C, D, E, F, G, H). 

The system transformed the participants’ age and the years of career into ordinal variables. The analysis of the correlations consisted of dividing these into five and four different classes (as described below in Table 1, Table 2, Table 3, Table 4, Table 5, Table 6, Table 7 and Table 8), respectively. The acceptance threshold for correlation values was higher than 0.60. R software [16] and the psych [18] and ggplot2 [19] packages were used to examine the data.

## 3. Results

Seven hundred and four physiotherapists replied to the online questionnaire. We deemed five questionnaires invalid due to insubstantial answers (non-existent universities/schools and/or disproportionate time taken to obtain the qualification). No questionnaire was incomplete. The remaining 699 questionnaires (99.2%) were deemed valid for the data analysis. 

### 3.1. Demographic Characteristics of Participants (SECTION A)

The majority of the participants (n = 385; 55%; 95% CI 55.09–55.23) are male with an average age of 32 years. The sample is composed of five five-year age groups (from 20 to 40) with the open final class (40+). We observe that the respondents belong mainly to the 26–30 group of age (36.82%), followed by the 31–35 group (20.20%). The youngest graduates are between 21 and 25 years old (17.34%). The two over-40s groups comprise 25.64% (36–40 and over 40). The average years of work experience are 8.7. A total of 36.96% (n = 258, 95% IC, 233.4–283.4) of respondents have less than 5 years of experience. A smaller number of respondents have 10–20 years of work experience (16.91%, n = 118, 95% IC 98.7–137.6). The number of respondents with more than 20 years of work experience is just as small (9.74% n = 68 95% IC, 52.7–83.5). A total of 70% (40% men and 30% women) of respondents work in private practice (n = 489/699, 95% IC 466–513.4). A total of 20% (n = 139, 95% CI 118.5–159.9) work in private facilities. Only 8% (n = 58, 95% CI 43.8–72.4), mostly female, work in a public facility. The remaining 1.72% (n = 12, 95% IC 5.3–18.8) work in universities with different job roles.

A total of 49.57% of the physiotherapists (n = 346, 95% IC 320–372.4) work in Northern Italy and 46.42% (n = 324, IC 298–350) graduated there as well. A total of 38.12% of the physiotherapists (n = 266, 95% CI 241.2–291.5) currently work in Southern Italy. Of those, 38.97% of them (n = 272, 95% IC 247.1–297.7) graduated in southern Italian regions. The remaining 12.32% (n = 86, 95% IC 69.1–103.2) work in central Italian regions, and 14.61% of them (n = 102, 95% IC 83.8–120.5) received their Bachelor in these regions.

### 3.2. Knowledge and Management of Contextual Factors (SECTION B)

Our team researched where the participants heard about CFs for the first time. A total of 33.95% (n = 237, 95% IC 212.8–261.9) reported having become aware of them during the 1st cycle degree program. The most frequently reported universities were: University of Ferrara (74%), Sacred Heart Catholic University of Rome (50%), University of Padua (47.7%), University of Genoa (47.2%).

The majority of the participants—40.83% (n = 285, 95% IC 259.9–310.9)—consider CFs “*any element voluntary and involuntary, with which the patient interacts during the treatment*”. A total of 16.33 % (n = 114, 95% IC 95–133.3) selected answers 1 and 2. A total of 9.89% (n = 69, 95% IC 53.6–84.6) selected answers 1 and 3.

All of the remaining nine combinations received a total number of answers equal to 15%, with the combination of the first three answers having 4.30% (n = 30, 95% IC 19.5–40.6). The remainder did not exceed 2.01% per response. (Figure 1).

A total of 82.9% (n = 583, 95% IC, 561.4–600.3) of the participants chose the representation of the CFs connected with the therapeutic relationship. A total of 75.8% (n = 533, 95% IC, 506.5–551) chose the therapeutic setting. Only 36.82% (n = 257, 95% IC, 232.4–282.4) chose all five answers in the representation of CFs (patient characteristics, physiotherapist characteristics, treatment characteristics, therapeutic setting, therapeutic relationship). The remaining 5.87% of the respondents chose the answers that include the therapeutic relationship and the therapeutic setting (n = 41) (Figure 2).

Most participants, 57.16% (n = 399, 95% IC 373–425.2), in clinical practice pay more attention to the therapeutic relationship.

The majority of the participants, 54.4% (n = 380, 95% IC 354.7–406.4), find it more useful to know how to manage contextual factors to improve the clinical response to physiotherapy treatment. A Likert scale examined the frequency of the use of CFs in clinical practice. Values ranged between “1” and “5”. “1” indicates a minimal use of CFs and “5” stands for greater use of CFs. The detected frequency seems to be relatively high, with an average of 3.7 (1095 DS).

### 3.3. Physiotherapist Characteristics (SECTION C)

A total of 76.93% (n = 537, 95% IC 515.9–559.6) of the physiotherapists lend great importance to communication strategies in order to achieve better therapeutic results. A total of 15.76% (n = 110, 95% CI 91.3–129) indicated that the best results arise from professional reputation.

The questionnaire examined the case in which a patient asks about the use of medical treatments, the effectiveness of which is without scientific evidence. A total of 50.29% (n = 351, 95% IC 325.6–377.4) of the sample offer a different treatment that is more suitable for the clinical situation.

### 3.4. Patient Characteristics (SECTION D)

Our team examined the physiotherapists’ method of working with respect to the patient’s experience with other treatments and different professionals. A total of 44.70% (n = 312, 95% IC 286.7–335.2) believe that previous experiences do not affect the treatment.

The participants indicated that they stimulate the patient’s positive expectations, aiming at the success of the therapeutic management (45.27%, n = 316, 95% IC 290.7–342.2). Otherwise, the physiotherapists take into account the patient’s expectations due to the influence these have on the outcome of the treatment (39.97%, n = 279, 95% CI 254–304.8). The survey analyzed the clinical approach according to the patient demographic characteristics. A total of 67.48% (n = 471, 95% IC 447.4–495.9) of the physiotherapists adapt their treatment depending on the patients’ age and gender.

### 3.5. Patient–Physiotherapist Relationship (SECTION E)

Physiotherapists attribute value to the relationship with the patient, with 57.88% (n = 404, 95% IC 379–430.2) of the participants corresponding with the patient during the treatment path. Next in order of importance were the communicative aspect, willingness to adapt to the patient’s demands, and the therapeutic touch. Concerning the real communication with the patient, the most significant element, identified by 36.25% (n = 253, 95% IC 228.5–278.3) of the physiotherapists, is active listening.

### 3.6. Treatment Characteristics (SECTION F)

With respect to the elements most taken into account during therapeutic planning, 54.87% (n = 383, 95% IC 357.8–409.3) of the participants believe it is essential to plan the sessions according to the patient’s clinical condition. Only 3.15% (n = 22, 95% IC 13–31.1) believe that group sessions are relevant for patients with similar problems. A total of 40.69% (n = 284, 95% IC 259–309.9) of respondents claim that the price of the service is standard. Only 2.44% (n = 17 95% IC 9–25) of the participants determine the cost depending on the prescribed brand’s therapy.

### 3.7. Therapeutic Environment (SECTION G)

Of the participants, 57.59% (n = 402, 95% IC 377–428.2) promote the therapeutic context by focusing on general comfort (natural lighting, ventilation, heating, etc.). The next most frequent elements are: environmental design, and next-gen equipment.

### 3.8. Conclusions (SECTION H)

For 96.42% (n = 673, 95% IC 664.3–683.6) of respondents, the contextual factors are significant and the study of their application should be deepened during training. Only 3.58% (n = 25, 95% IC 15.4–34.7) do not consider contextual factors relevant.

A comprehensive analysis of the questionnaire answers is provided in detail in Table 1, Table 2, Table 3, Table 4, Table 5, Table 6, Table 7 and Table 8.

## 4. Tables and Schemes

**Table 1 healthcare-09-00689-t001:** **Participant’s characteristics** (n = 699). %: percentage, n: number of participants, 95% IC: 95% confidence interval.

Gender	Values	95% CI
Men, n (%)	385 (55.0)	55.09–55.23
Women, n (%)	314 (45.0)	287.7–339.2
**Years, average (SD)**		
21–25, n (%)	121 (17.34)	
26–30, n (%)	257 (36.82)	
31–35, n (%)	141 (20.20)	
36+, n (%)	179 (25.64)	
**Job Title**		
Freelancer, n (%)	489 (70)	466–513.4
Private facility n (%)	139 (20)	118.5–159.9
Public facility, n (%)	58 (8)	43.8–72.4
University professor, n (%)	1 (0.1)	0–3
Post-graduate studies (master) professor n (%)	8 (1.5)	2.5–13.5
Private courses or studies post-graduate professor, n (%)	3 (0.4)	0–6.4
Researcher, n (%)	0 (0)	0
**Work field**		
Musculoskeletal, n (%)	507 (72.64)	484.6–530.8
Neurological, n (%)	73 (10.46)	57.2–89
Geriatric, n (%)	63 (9.03)	48.2–77.9
Sports n (%)	33 (4.7)	22–44
Cardiorespiratory, n (%)	22 (3.15)	13–31.1
None of the above n (%)	0 (0)	0
**Working Italian region**		
North, n (%)	346 (49.57)	320–372.4
Center, n (%)	86 (12.32)	69.1–103.2
South, n (%)	266 (38.12)	241.2–291.5
**Participants’ graduation**		
**Italian region**		
North, n (%)	324 (46.42)	298–350
Center, n (%)	102 (14.61)	83.8–120.5
South, n (%)	272 (38.97)	247.1–297.7
**University**		
**North, (%)**	46.42	
**Center, (%)**	14.61	
**South, (%)**	38.97	

**Table 2 healthcare-09-00689-t002:** **Contextual factors (CFs).** %: percentage, n: number of participants, 95% IC: 95% confidence interval.

Environment in Which You Heard of CFs for the First Time	Values	95% CI
Bachelor’s degree, n (%)	237 (33.95)	212.8–261.9
Master, n (%)	190 (27.22)	167.2–213.3
Never heard	111 (15.90)	92.2–130.1
about CFs, n (%)		85.7–122.6
Private course, n (%)	104 (14.90)	32.3–57.8
Social media, n (%)	45 (6.45)	4.6–17.5
Magistral degree, n (%)	11 (1.58)	
**What the CFs are**		
Any element, voluntary or involuntary, with which the patient interacts during treatment, n (%)	536 (76.2)	514.9–558.6
Specific therapeutic tool able to influence the treatment outcome through neurophysiological mechanisms, n (%)	239 (34)	214.8–263.9
Intervention without a specific effect but with a possible non-specific effect, n (%)	140 (19.9)	119.5–161
Diagnostic tool capable of distinguishing between a psychological problem and an organic problem, n (%)	51(7.3)	37.6–64.6
I don’t know how to describe it, n (%)	68 (9.74)	52.7–83.5
The **CF considered most important**		
Patient characteristics, n (%)	168 (24.07)	146.1–190.4
Physiotherapist characteristics, n (%)	20 (2.87)	11.4–28.7
Treatment characteristics, n (%)	84 (12.03)	67.3–101
Therapeutic setting, n (%)	27 (2.87)	17–37
Therapeutic relationship, n (%)	399 (57.16)	373–425.2
**Usefulness of their knowledge**		
Improve the therapeutic relationship, n (%)	144 (20.63)	123.2–165.2
Improve the clinical response to physiotherapy treatment, n (%)	380 (54.4)	354.7–406.4
Improve the patient’s satisfaction, n (%)	166 (23.78)	144.2–188.3
Control symptoms, n (%)	8 (1.15)	2.5–13.5

**Table 3 healthcare-09-00689-t003:** Frequency of use CFs.

Frequency of use of CFs	Likert Score Average	1 (Never) n (%), 95% IC	2 (Few) n (%), 95% IC	3 (Sometimes) n (%), 95% IC	4 (Often) n (%), 95% IC	5 (Always) n (%), 95% IC
	3.7	34. (4.87). 22.9–45.2	45. (6.45). 32.3–57.8	171. (24.5). 149–193.5	236. (33.81). 211.8–260.9	212. (30.37). 188.5–236.1

**Table 4 healthcare-09-00689-t004:** **Physiotherapist characteristics**, %: percentage, n: number of participants, 95% IC: 95% confidence interval.

Most Important Professional Personal Element	Values	95% CI
Professional reputation n (%)	110 (15.76)	91.3–129
Uniform n (%)	9 (1.29)	3.2–14.9
Hygiene and cleanliness, n (%)	35 (5.01)	23.7–46.4
Communication strategies, n (%)	537 (76.93)	515.9–559.6
Charge, n (%)	7 (1.00)	1.8–12.2
**Strategy adopted in front of a request for treatment not scientifically proven**		
Provide the requested therapy by the patient while aware of its ineffectiveness, n (%)	52 (7.45)	38.5–66.7
Inform the patient about inappropriate therapy; n (%)	224 (32.09)	200.1–248
Suggest a different treatment that reflects your beliefs, knowledge and experience about the clinical picture; n (%)	351 (50.29)	325.6–377.4
Suggest the requested therapy, but only after the previous provided therapy fails, n (%)	71 (10.17)	55.4–86.8

**Table 5 healthcare-09-00689-t005:** **Patient characteristics,** %: percentage, n: number of participants, 95% IC: 95% confidence interval.

Influence of Past Experiences on the Therapeutic Strategy	Values	95% CI
Perform the therapeutic strategy adopted by the previous healthcare professional because it is beneficial to the patient, even if not supported by scientific evidence, n (%)	104 (14.90)	85.7–122.6
Do not perform therapy deemed most appropriate and suitable for the patient if he reports negative experiences in the past, n (%)	245 (35.10)	220.06–270.01
Previous experiences do not affect the choice of the treatment at that the time, n (%)	312 (44.70)	286.7–335.2
Always comply with my patient’s request n (%)	37 (5.30)	25.4–48.7
**Management strategy and patient expectations**		
The patient’s expectations do not influence the treatment, n (%)	25 (3.58)	15.4–34.7
Take into account the patient expectations because they may affect the treatment outcome regardless of the specificity of it, n (%)	279 (39.97)	254–304.8
Always try to stimulate positive expectations in order to strengthen motivation, therapeutic alliance and clinical outcome, n (%)	316 (45.27)	290.7–342.2
Try to mitigate negative expectations (based on information gaps) by explaining, before the treatment, the proven effectiveness of the chosen strategy, n (%)	78 (11.17)	61.8–94.4
**Management strategy and patient’s age and gender**		
Always adapted clinical practice to patient’s age and gender	471 (67.48)	447.4–495.9
Patient’s gender and age does not influence the treatment choice, n (%)	17 (2.44)	9–25
Believe it is necessary to make a distinction but I do not know how to adapt my clinical practice, n (%)	51 (7.31)	37.6–64.6
Different approach depending on patient’s age, but I do not believe gender is relevant, n (%)	159 (22.78)	137.5–181

**Table 6 healthcare-09-00689-t006:** **Patient–physiotherapist relationship**. %: percentage, n: number of participants, 95% IC: 95% confidence interval.

Most Important Relational Element	Values	95% IC
Communication, n (%)	215 (30.8)	191.4–239.2
Availability in front of the patient’s requests, n (%)	47 (6.67)	34.1–60.1
Patient-centered care pathway, n (%)	404 (57.88)	379–430.2
Therapeutic touch, n (%)	32 (4.58)	21.2–42.9
**Key element of communication**		
Active listening, n (%)	253 (36.25)	228.5–278.3
Use of technical speech, n (%)	4 (0.57)	0.1–7.9
Verbal expressions of support and encouragement, n (%)	62 (8.8)	47.3–76.8
Humor and sympathy, n (%)	28 (3.8)	17–37
Paraphrases, images and metaphors to help the patient understand his medical condition, n (%)	176 (25.21)	153.7–198.8
Explanation of the effects and treatment execution (overt therapy), n (%)	97 (13.9)	79.2–115.1
Consistency between verbal, paraverbal and non-verbal language, n (%)	79 (11.30)	62.7–95.5

**Table 7 healthcare-09-00689-t007:** **Treatment characteristics.** %: percentage, n: number of participants, 95% IC: 95% confidence interval.

Elements Taken into Account during Planning	Values	95% IC
Priority for one-to-one session, n (%)	232 (33.24)	207.9–256.7
Group session for patients with similar problems, n (%)	22 (3.15)	13–31.1
Patient’s availability, n (%)	61 (8.74)	46.5–75.5
Patient’s clinical status, n (%)	383 (54.87)	357.8–409.3
**Price of treatments**		
Better branded therapies have a higher cost, n (%)	17 (2.44)	9–25
The price depends on the time spent during the session, n (%)	114 (16.33)	95–133.3
The price depends on the treatment type (innovations, advanced technology, etc.), n (%)	125 (17.91)	105.3–145
The price depends on the treatment complexity, n (%)	119 (17.05)	99.7–138.7
The price depends on the clinician experience, n (%)	39 (5.59)	27.2–51
The treatment price is standard, n (%)	284 (40.69)	259–309.9

**Table 8 healthcare-09-00689-t008:** **Therapeutic environment.** %: percentage, n: number of participants, 95% IC: 95% confidence interval.

Therapeutic Environment Care	Values	95 % IC
Setting design, n (%)	172 (24.64)	149.9–194.6
Technological equipment, n (%)	31 (4.44)	20.4–41.7
Architecture to respect privacy, n (%)	93 (32)	75.5–110.7
General comfort, n (%)	402 (57.59)	377–428.2

## 5. Discussion

This survey is the first national one researching Italian physiotherapists’ knowledge and use of CFs. The main achievement of this study is to prove that participants are aware of context-related effects in their clinical practice. These effects can be triggered by specific characteristics of the intervention provided and by non-specific characteristics not inherent to it. However, some potentially significant aspects are still underestimated. According to our research, most Italian physiotherapists deem CFs to be any element, even unintentional ones, with which the patient interacts during treatment. In previous OMPT surveys [5,20], instead, the results revealed that physiotherapists regarded them as random elements. Few considered them to be a specific therapeutic tool capable of influencing patients’ clinical outcomes [5,9,11]. This point of view may be related to limited knowledge regarding the neurophysiological mechanisms behind the therapeutic effects of CFs [1,21,22]. Several participants stated having learned about CFs during a 2nd cycle degree class. The lower attendance among the participants of 2nd cycle degree courses may have affected this response. This may depend on how the Italian institutional system configures physiotherapists’ career path. In line with most studies, Italian physiotherapists identify the contextual factor “therapeutic relationship” as the essential element for fair clinical success [9]. While it is true that the therapeutic alliance is known as an outcome predictor during musculoskeletal physical therapy [23], this result does not justify the small percentage of participants who believe it is significant to improve the “characteristics of the physiotherapist” in order to achieve the same result. It is clear now that the patient’s perception of the physiotherapist—their qualifications and reputation, the way they dress—may influence the overall perceptions of the patient [9,11]. In detail, by questioning the participants about the specific tools available to them (aiming to increase the probability of success), we came across communication strategies, followed by professional reputation. Professional reputation was the element that received the second-most feedback, unlike other surveys on the same topics [7]. However, the majority of the sample barely use this CF and do not value it to be significant. This may be due to a particular lack of knowledge of its clinical relevance and the complex concept of professional identity [5]. This is also confirmed by the low percentage of participants who opted for the use of a uniform as a feature for enhancing the therapeutic result.

We observed that, according to different age groups, the usage of CFs varies. Most respondents belonging to the 26–30 age group selected the answer “physiotherapist–patient relationship” as a first choice. The 36–40-year-old group pays attention more to the patient’s characteristics. The clinicians’ age relative to their years of work shows that experience leads clinicians to pay more attention to the patient than to their relationship with them. They take into account the patient’s overall characteristics in many ways: gender, age, expectations and previous experiences. In situations where a patient desires a treatment whose efficacy lacks available scientific evidence, half of the respondents suggest a different treatment that is more suitable for the clinical picture. This specific question underlines a hotly debated topic: providing the best therapy in terms of scientific evidence without forgetting internal elements (i.e., expectations and previous experiences) that may have a significant effect [2,9,11,24,25]. As suggested by the current literature, physiotherapists who decide to advise a different treatment, regardless of the patient’s demands, should carefully balance the positive characteristics of medical treatment with the negative ones of the other [26]. Behind this, there is a significant aspect of the therapeutic relationship: the sharing of decision making [27,28,29]. If there are two treatment options, one desired by the patient and one deemed most appropriate by the therapist, the shared decision-making process could change the patient’s belief. The patient transitions from uninformed preferences to better-explained preferences [27,28,30]. Physiotherapists that prescribe a treatment without scientific proof and do not inform their patients are adopting a non-transparent and misleading communication. This disrespects the ethical principles behind a prescribed therapy (i.e., the principle of autonomy; informed consent) [27,28,30].

The patient’s past experiences with previous healthcare professionals are not considered to be a factor influencing the therapeutic choice pursued. This result is very different from current evidence showing that associative learning and past experiences lead to mechanisms of action underlying the nocebo and placebo effect [9,29,31]. Redundant therapy associations with some of its characteristics (i.e., color, smell, shape, invasiveness, etc.) and associations with verbal messages that arouse expectations (i.e., this therapy will decrease your pain) may activate context-related effects in healthy people and patients [11,32,33]. The spread of positive expectations can modify subjective results (i.e., pain and anxiety) and objective results (feature and disability) in patients suffering from musculoskeletal pain [32,34].

The Italian physiotherapists, when questioned on the management of the demographic characteristics, claim to take into account age, but do not always consider gender. This variable plays a significant role in clinical practice [35,36]. The patient may experience pain or behave differently in response to it. Gender and age can also affect how people interpret pain or the behavior of other people [37,38,39]. Women tend to show and experience more pain than men. Men, instead, consider themselves more determined and more focused on objectives [37]. Since men value themselves as being more determined, it may suggest that when it comes to making decisions about diagnosis or case management, physiotherapists should rely more on these aspects. Men seem to value result-oriented communication. By contrast, too much determination may come across as paternalistic or not very sympathetic for the female gender [40]. Understanding these characteristics could be useful for improving interactions with patients and generating future lines of research for those who believe that these factors are relevant, but do not know how to develop them or, on the contrary, for those who believe that gender has no impact on treatment choice.

Regarding the “treatment characteristics”, the patient’s engagement with their treatment path is considered by participating physiotherapists to be a result of improved self-efficacy and self-responsibility in patients with musculoskeletal pain [9,35,41]. Therapeutic touch, on the other hand, is a relational element that is rarely taken into account. A profession that spends such a large amount of time with patients should understand that it is possible to deliver positive and negative messages, specifically through touch [11,42]. The participants also believe that the explanation of treatment effects and performance (overt therapy) are less influential CFs. Physiotherapists have little practical knowledge of their clinical relevance. Different surveys on the same topic indicate that patients greatly appreciate this element, and therefore clinicians should take it into consideration more often [7]. 

Overall, our results suggest to physiotherapists the need to consider communication-specific CFs as triggers of placebo and nocebo effects capable of having a considerable impact on patient outcomes, in accordance with the evidence reported in the fields of medicine and physiotherapy [22,43]. Seeing how physiotherapists plan the therapy, it seems that Italians tend to underestimate the effect of positive social interactions. Positive interactions could arise from contact with patients with a similar medical history. On the contrary, they prefer to choose one-to-one sessions. The literature shows how patients who support each other’s efforts to improve the health make the environment more motivating. They also develop supportive relationships and emotional support [44,45]. With respect to the reasons for the price of the service provided, there was a difference between the different age groups. The 20–36 age group claimed that the price of the session depends on the time spent. Other age groups reported a standard price, regardless of different recommended variables (i.e., brand, complexity, experience). The current literature does not include articles about the analysis of this field, but we know it is one of the factors influencing patients’ choices, and further investigation is required to study this aspect and to validate the investigation.

Italian physiotherapists aim to enhance the therapeutic context by taking care of general comfort (natural lighting, ventilation, heating). There are fewer answers on environmental design, appropriate architecture that respects privacy, and on the use of next-gen equipment. In clinical practice, regular use of CFs (i.e., relaxing music, soft light, comfort) with the best evidence-based treatment offers physiotherapists the opportunity to manage patients’ symptoms; fear, avoidance, and anxiety are commonly related to musculoskeletal pain [11]. Current design patterns emphasize the therapeutic effects of appropriate design, what Rehn Schuster identified as the “placebo design effect” [46].

Almost all participants unanimously answered willingly to apply this topic further during practical training. This shows that even physiotherapists acknowledge the importance of this topic, which has developed in the literature over the last decade among physicians, as well as physiotherapy and nursing students [5,6,8,46].

### Strengths and Weaknesses of the Study

The methodological choice was a strong point, and is regarded as a useful tool for grasping the opinions and the perspectives of a large sample of healthcare workers. Other surveys have applied this methodological approach for placebo and nocebo effects and CFs [4,7,8,25]. This allowed us to extend our analysis. For the first time, a team of researchers individually analyzed each CF that is currently recognized. 

The questions about Italian regions and universities of origin made it possible to draw up reports from a demographic point of view, seeking a future perspective for education. 

A large number of answers came from private practice physiotherapists specialized in orthopedics (70.06%). Their responses may not apply to physiotherapists specialized in other areas (e.g., neurological) and/or used in different contexts (e.g., hospitals). We deem this to be a point of weakness in the research. However, it may suggest future studies in this field. Furthermore, the way participants were asked how to define CFs with a closed question with four answer options could have given some preliminary insights to the participant, thus influencing their knowledge of the subject.

## 6. Conclusions

The responses to this survey show that Italian physiotherapists believe that the perception of pain and the success of their treatment also derive from the presence of positive CFs; however, the discrepancy between some answers shows the uncertainty of their conscious use, perhaps due to the complexity of the phenomenon. Most Italian physiotherapists assert that they do not use CFs on a daily basis, preferring verbal communication and a patient-centered approach. Otherwise, little attention is paid to the characteristics of the physiotherapist and to socially adequate environments to trigger placebo and nocebo effects. The research was useful in underlining the more widespread implementation of CFs in the musculoskeletal field, but less so for the other specializations. Future research should provide information on the difficulties that may hinder its adoption. The results of the research show how much it is necessary to enrich undergraduate and post-graduate study programs with these themes, to ensure the adequate use of the context. New implementation strategies must be brought into training courses so that as a student, it must be clear that the final result is always the combination of the specific effect of the treatment and the neurobiological effects caused by the variables related to the context. Future qualitative research should investigate the relationships between the determinants of patient satisfaction and expand the investigation into other physiotherapy settings to provide a thorough understanding of this topic.

## Figures and Tables

**Figure 1 healthcare-09-00689-f001:**
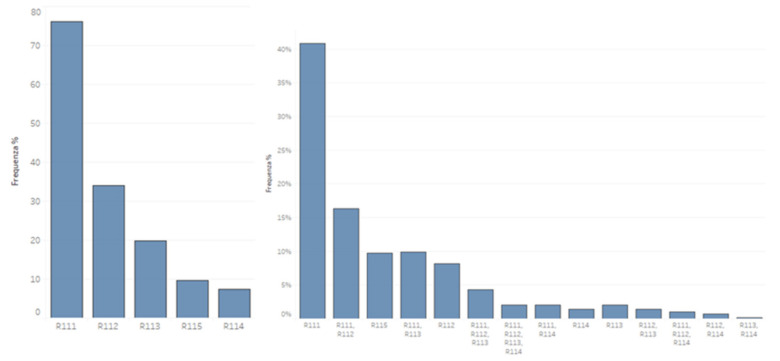
Single and multiple answers. % percentages of module 1 response count for each D1,D11: What are CFs; (R111) any element, voluntary and involuntary, with which the patient interacts during the treatment; (R112) specific therapeutic tool that influences the outcome of the treatment through neurophysiological mechanisms; (R113) intervention without a specific effect but with a possible non-specific effect; (R114) diagnostic tool that distinguishes a psychological-type problem from an organic-type problem; (R115) I can’t define it.

**Figure 2 healthcare-09-00689-f002:**
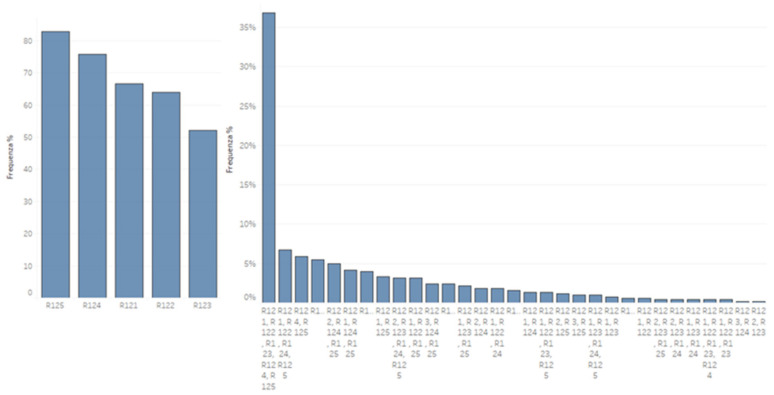
Single and multiple answers. % percentages of module 1 response count for each D1, D12: Which of these do you think represents a CF? (R121) patient characteristics; (R122) physiotherapist characteristics; (R123) treatment characteristics; (R124) Therapeutic environment; (R125) patient–physiotherapist relationship.

## Data Availability

Not applicable.

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
