# Peer review of "Understanding Contextual Factors Effects and Their Implications for Italian Physiotherapists: Findings from a National Cross-Sectional Study"

_healthcare, 2021, doi:10.3390/healthcare9060689_

Round 1

Reviewer 1 Report

Dear authors,

I congratulate you for the research carried out because I believe that it is important in patient care and there are not many studies in relation to contextual factors.

However, I believe that some modifications are necessary to improve the manuscript.

  1. Title.

Indicate in the title the type of research carried out.

  1. Abstract.

Indicate the research objective in the abstract and the type of methodology chosen.

  1. Introduction.

Line 62- This statement is from the authors of the article or is it from reference 5 ?, because it seems that reference 5 only talks about its use and this seems more like an opinion of the authors.

Moreover, it is part of the research objective, therefore it should be withdrawn from this section or raised as a hypothesis.

  1. Material and methods.

Line 80-84 Was there a heterogeneous group of specialties or were there differences between whether your job was public or private? It must be specified here although they will also do it later.

Line 87-What bibliography did you consult? Did they do a search? What kind of search? With what inclusion and exclusion criteria? And what keywords.

Tell what professional profile these authors have, are they physiotherapists, and for what group or in which study did they ask these questions?

Line 95-97-Cite which are those previous questionnaires and put your reference

Line 104-This procedure for making and choosing questions, why was it decided to do it this way, do you have evidence that other studies did it this way? If so, quote them.

Line 123-It is necessary to publish the questionnaire as an annex to the manuscript. It is the way to validate the questionnaire and the way to be able to make translations into other languages, and ultimately to advance the research and help patients.

  1. Results.

It would be appropriate to try to make the results more visibly attractive and easy to print with more graphics.

&. Discussion.

It is important to try to give the information in a more summarized way in this section.

Line 415 It would be interesting to propose future lines of research. Where you comment that it is carried out in other professions and it would also be convenient to validate the survey.

Line 444 is a section that should be included in the discussion. It should be added that the fact that patients are not asked about the influence of FC may be a bias. The final is a two-part relationship that must be listened to in order to draw conclusions about which CFs are most influential and which ones they need the most or value the most for their health.

  1. Conclusions.

They should be more specific, short and concise.

Author Response

Response to Reviewer 1 Comments

Dear reviewer, thanks for your interest about contextual factors and our study; thanks for your indications to improve the manuscript.

Point 1: Title.

Indicate in the title the type of research carried out.

Response 1: “cross-sectional study”

Point 2: Abstract.

Indicate the research objective in the abstract and the type of methodology chosen

Response 2: Add in line 16-18

Point 3 Line 62- This statement is from the authors of the article or is it from reference 5 ?, because it seems that reference 5 only talks about its use and this seems more like an opinion of the authors.

Moreover, it is part of the research objective, therefore it should be withdrawn from this section or raised as a hypothesis

Response 3: This statement is from the authors, we raised as an our hypothesis line 64.

Point 4 Material and methods.

Line 80-84 Was there a heterogeneous group of specialties or were there differences between whether your job was public or private? It must be specified here although they will also do it later.

Line 87-What bibliography did you consult? Did they do a search? What kind of search? With what inclusion and exclusion criteria? And what keywords.

Tell what professional profile these authors have, are they physiotherapists, and for what group or in which study did they ask these questions?

Line 95-97-Cite which are those previous questionnaires and put your reference

Line 104-This procedure for making and choosing questions, why was it decided to do it this way, do you have evidence that other studies did it this way? If so, quote them.

Line 123-It is necessary to publish the questionnaire as an annex to the manuscript. It is the way to validate the questionnaire and the way to be able to make translations into other languages, and ultimately to advance the research and help patients.

Response 4: Line 93-97 to specify the group and the job.

For other authors we decide to include only the Rossetini’s study in bibliography.

Previous questionnaire are indicate in line 108 (bibliography 5-8)

We follow DeLeeuw (line 116)

We can attach the questionnaire

 Point 5 It would be appropriate to try to make the results more visibly attractive and easy to print with more graphics.

Response 5: Figures at the end: line 528 and line 534

Point 6 It is important to try to give the information in a more summarized way in this section.

Line 415 It would be interesting to propose future lines of research. Where you comment that it is carried out in other professions and it would also be convenient to validate the survey.

Line 444 is a section that should be included in the discussion. It should be added that the fact that patients are not asked about the influence of FC may be a bias. The final is a two-part relationship that must be listened to in order to draw conclusions about which CFs are most influential and which ones they need the most or value the most for their health.

Response 6: we cut and summarize the text

Point 7

Response 7: write again in a different way line 408-425

Reviewer 2 Report

This study is a very interesting study that surveyed Italian physiotherapists on the effectiveness of contextual factors. It suggests that contextual factors can be used effectively in various physical treatments in the future.

However, in order to improve the overall quality of the paper, I recommend to revise the following.  

-At first, define 'contextual factor' clearly in introduction part.

-Insert literature review part between Introduction and Materials and Methods.

-Parts 3.1 to 3.7 are written by verbally too long. Instead use tables or figures for better readability.

-Also, overall parts of chapter 5(Discussion) can be organized with tables or figures. 

-The  conclusion is rather vaguely expressed. The clear conclusion must be presented.

Author Response

Response to Reviewer 2 Comments

Dear reviewer, thanks for your interest about contextual factors and our study; thanks for your indications to improve the manuscript.

Point 1: At first, define 'contextual factor' clearly in introduction part.

Response 1: line 45-46

Point 2: Insert literature review part between Introduction and Materials and Methods.

Response 2: little section for a narrative review line 76-84

Point 3 Parts 3.1 to 3.7 are written by verbally too long. Instead use tables or figures for better readability.

Response 3: We summarized this section

Point 4 Also, overall parts of chapter 5(Discussion) can be organized with tables or figures. 

Response 4: figures at lines 528 and 534.

 Point 5 The conclusion is rather vaguely expressed. The clear conclusion must be presented

 Response 5: Write again in a more specific way line 408-425